# Less Is More? Field Evaluation of Short-Stature Banana Cultivars in a Mediterranean Environment

Idan Elingold [1], Avishi Londener [1], Amit Rimmer [1], Avi Tsarfaty [1], Gilad Hadar [1] and Navot 'Galpaz [2,*]

[1] Jordan Valley Banana Research Station, Zemach 15132, Israel; idane@zemach.co.il (I.E.); avishke@gmail.com (A.L.); rimmeramit@gmail.com (A.R.); avit@zemach.co.il (A.T.); giladhadar4@gmail.com (G.H.)

[2] Northern R&D, MIGAL-Galilee Research Institute, Kiryat Shmona 11016, Israel

* Correspondence: navotg@migal.org.il

**Abstract:** The shift to cultivation of banana ('Musa (AAA) Cavendish subgroup') in screenhouses in Israel has resulted in a significant increase in plant height and, consequently, increased maintenance costs. Here, we evaluated the genetic approach to reducing plantation height. Advanced selections of the local cultivars 'Adi', 'Zelig', and 'Gal', selected for reduced height, were evaluated in the field. Growth and yield parameters were recorded and compared with the industry standard cv. 'Grand Naine' for four crop cycles. 'Adi' and 'Zelig' were shorter than 'Grand Naine', by 20% and 10%, respectively, whereas 'Gal' lost its short stature over the years. In addition, leaf area was reduced in the low-stature cultivars. Cumulative yield of 'Adi' and 'Zelig' was higher than that of 'Grand Naine', by 8.8% for 'Adi' and 5.0% for 'Zelig', due to higher plant density and number of harvested bunches. This multiyear study highlights 'Adi' and 'Zelig' as short-stature, highly productive cultivars with the potential for improved water-use efficiency.

**Keywords:** dwarf mutants; cultivar trial; yield; Musa (AAA) Cavendish subgroup

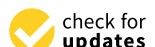



## 1. Introduction

Banana (*Musa* spp.) is ranked first in global production among fruit crops, with 114 million tons produced in 2017 [1]. Banana is also the largest fruit crop in Israel in terms of production, with ca. 180,000 tons produced in a 3100 ha cultivation area in 2019 [2]. Local selections of cv. 'Grand Naine' (genome AAA) are responsible for ca. 95% of banana production in Israel, due to their relatively short crop cycle, high fruit quality, and high yields under the limiting subtropical environmental conditions [2,3].

The large and thin banana leaves are prone to shredding by wind, reducing photosynthetic efficiency and, consequently, productivity [4]. To reduce wind damage, at the beginning of the century the Israeli banana industry shifted cultivation from open fields to screenhouses. Inside the screenhouses, the plants became healthier and more robust, and productivity improved by 30% due to higher bunch weight and a drop in bunch loss. Additional benefits included ca. 25% water savings due to net shading, which significantly reduced evapotranspiration, the ability to grow in plots exposed to high winds that were not suitable for open-field cultivation, improved fruit quality, and prolongation of the harvest period [5].

However, the transition to cultivation inside screenhouses has one major downside: the robust plants are about 50 cm taller than those grown in the open field; plant leaf area and leaf area index are higher, resulting in higher co-shading [5,6]. The excessive plant height also results in additional production costs: the need to build high (reaching 6 m) and expensive screenhouses, and extra labor costs for plantation maintenance, especially bunch-related actions: propping, bagging, dehanding, and harvesting. Moreover, to reduce overshading by the bigger plants growing inside the screenhouse, planting density must be reduced, resulting in a lower number of harvested bunches per unit area. Therefore,

a significant reduction in plant height and canopy area is urgently needed in the Israeli banana plantations. Common effective agrotechnical practices used for height and size control in other fruit crops are not available for banana: grafting—and therefore the use of dwarfing rootstocks—is challenging in monocots due to the lack of cambium tissue [7]. Topping, a common practice for height and size control in woody plantations and orchards, is not realistic in plantations of the herbaceous banana, and the application of growth retardants results in the reduced size of all organs, including bunches, probably due to the determinate growth habit of the banana ([8], Galpaz N, unpublished data).

In contrast to indeterminate plants, in which competition for assimilate allocation can occur between the vegetative and reproductive phases, in banana, as a determinate plant, there is normally a strong positive correlation between plant vigor and bunch size, and pseudostem circumference and height are good predictors of bunch weight [9,10]. This correlation holds true at the cultivar level. However, the strong correlation between vegetative and reproductive traits are sometimes uncoupled at the inter-cultivar level, and short-stature mutants bearing bunches with similar weight to the taller parent plants have been selected and have served for the development of short-stature high-yielding cultivars, e.g., dwarf mutants of cvs. 'Gros Michel' and 'Gruesa Palmera' [11,12].

Along with the introduction of agrotechnical innovations, the development of new banana cultivars is a major driving force for improved productivity and profitability of the banana industry worldwide. Field trials with banana cultivars have been reported over the years in many countries, e.g., the Canary Islands, Turkey, Puerto Rico, Australia, and South Africa [9–11,13–16] to identify cultivars with superior performance under local conditions.

Most of the commercial banana cultivars dominating global production, including 'Williams' and 'Grand Naine', are triploid and sterile [17]. Therefore, conventional breeding through crosses is not a common practice in bananas. The biotechnological approach—genetic engineering—is a promising tool for precise banana breeding [18] but is currently performed only on a limited scale due to technological complexities and regulations, as well as public acceptance issues. Most of the genetic variations generated during in vitro propagation—designated somaclonal variations—result in inferior horticultural performance. However, banana breeders have harnessed somaclonal variation as an alternative approach to classical breeding by deliberately increasing somaclonal variation frequencies and screening for rare somaclonal variants that exhibit desirable traits, such as disease tolerance [19] and improved fruit quality [20]. Thus, over the years, short-stature and high-yielding somaclonal variants have been selected in Israeli banana plantations. The most promising ones were subjected to a continuous process of clonal evaluation and genetic improvement, through several selection cycles of individual plants showing improved performance, resulting in the release of short-stature, high-yielding selections. In the present study, the horticultural performance of three advanced selections of cvs. 'Adi', 'Zelig', and 'Gal' was compared with the industry standard, cv. 'Grand Naine'. 'Gal', selected in 1977, and 'Zelig', selected in 1980, are variants of 'Williams' and 'Grand Naine', respectively. Their major characteristics were low-to-moderate stature and heavy and compact bunches. 'Adi', selected at the beginning of the 2000s, is a Cavendish variant, also bearing a compact and heavy bunch [21]. These three cultivars are currently being grown in Israel on a limited scale. The purpose of this study was to evaluate the horticultural performance of advanced selections of these cultivars as a means of lowering screenhouse height and reducing labor costs.

## 2. Materials and Methods

### 2.1. Plant Materials

Four advanced selections of cvs. 'Grand Naine', 'Adi' (Selection no. 9108), 'Gal', and 'Zelig' were evaluated in this study. Tissue-cultured plants were obtained from Rahan Meristem Ltd. (Rosh Hanikra, Israel) and were then hardened for 12 weeks at the hardening facility of Kibbutz Kinneret before planting.

### 2.2. Experimental Design

The study was conducted between August 2015 and July 2020, in a 6 m high screen-house plot in Kibbutz Kinneret (32°70′ N, 35°59′ E; 200 m below sea level), located in the Jordan Valley, Northern Israel. Net type was polyethylene 'Cristal-Leno' 10% shading (Ginegar Agro, Ginegar, Israel). Tissue-cultured plants 40–50 cm in height and grown in 1.2 L pots were planted on 18 August 2015. Plant spacing was 3 m between mats along rows and 3.5 m between rows, with 2 or 3 plants per mat, alternately. Plant density was 2380 plants/ha. The experiment was conducted following a randomized complete block design, with six replicates. In each replicate, data were collected from 20 plants per cultivar, for an overall 120 plants per cultivar.

### 2.3. Plantation Cultivation

The plantation was subjected to standard agronomic practices in the Jordan Valley, Israel, which included the incorporation of 200 $m^3$/ha cattle manure and 1200 kg/ha super-phosphate into the soil before planting and nitrogen (300 kg N/ha per year as $NH_4NO_3$), potassium (450 kg K/ha per year as $K_2O$), and phosphorus (50 kg P/ha per year as $P_2O_5$) application via a drip irrigation system. The plantation was drip irrigated daily during the dry period (from April to November), for a total annual amount of ca. 1700 mm. Cultivation was almost chemical-free, with one annual application of 1 L/ha neonicotinoid (Confidor) in the autumn through the irrigation system.

Soil physical properties and mineral content in the research plot were typical of the Jordan Valley (Table S1). The climate in the Jordan Valley is characterized by two distinct periods: rainy and relatively cold (November–March) and dry and hot (April–October). Meteorological data, collected for the years 2000–2020, are presented in Table S2.

### 2.4. Data Collection

Data were collected over four crop cycles: parent crop (PC) and the following three ratoon cycles (R1–R3). Recorded parameters were: pseudostem height and circumference at 1 m after bunch emergence; bunch emergence date and harvest date; number of hands per bunch and bunch weight; and finger weight of the middle finger from the outer whorl of the third basal hand. Yield was calculated from the sum of the gross weight of all bunches harvested from each experimental plot. In Israel, where most of the fruit is marketed as whole bunches, this gives a close approximation of commercial yield. Maximal length and width of the third and sixth youngest leaves were recorded for 10 plants with bunches that emerged between 20 September 2020 and 30 September 2020. Leaf area per plant was calculated as: leaf length × leaf width × 0.82.

### 2.5. Data Analysis

All data were subjected to analysis of variance (ANOVA) through JMP version 10 (SAS Institute, Cary, NC, USA). When the variance was significant, the treatment means were grouped and analyzed by Tukey test at 5% error probability. Correlations between plant height and bunch weight were calculated using the multivariate function in JMP. In light of the high variability in the measured traits between the various crop cycles, each of the cycles was independently statistically analyzed.

## 3. Results

### 3.1. Growth Parameters

Cvs. 'Adi' and 'Zelig' were consistently significantly shorter than 'Grand Naine'. 'Adi' was the shortest among the tested cultivars. Its average pseudostem height over the four crop cycles was 57 cm lower than that of 'Grand Naine', whereas 'Zelig' was shorter by 27 cm than the industry standard cultivar (Table 1). 'Gal' was shorter than 'Grand Naine' in the PC and R3 cycles, but in the other two cycles it was taller, and on average, it was taller by 1 cm (Table 1). The average pseudostem circumference of 'Adi' was very similar to that of 'Grand Naine' (Table 1). Trends of increased and decreased pseudostem circumference

were monitored for 'Gal' and 'Zelig' (Table 1). 'Adi' was the sturdiest cultivar, with the lowest four-cycle average pseudostem height-to-circumference ratio (Table 1), resulting from a combination of short and wide pseudostem. The height-to-circumference ratio of cv. 'Zelig' was similar to those of the taller cultivars 'Gal' and 'Grand Naine', due to intermediate height and low pseudostem circumference (Table 1).

**Table 1.** Growth parameters for the various cultivars in each crop cycle and average of all cycles ($n = 60$ plants/cultivar for pseudostem height and circumference and 10 plants/cultivar for third-youngest leaf dimensions and area). AVG is the mean of the four cycles. Values in a row followed by the same letter are not significantly different ($p > 0.05$).

| | **Pseudostem Height (cm)** | | | | | **Pseudostem Circumference (cm)** | | | |
|---|---|---|---|---|---|---|---|---|---|
| **Cycle** | **Grand Naine** | **Gal** | **Zelig** | **Adi** | **Cycle** | **Grand Naine** | **Gal** | **Zelig** | **Adi** |
| **PC** | 290 a | 283 ab | 266 b | 238 c | **PC** | 68.5 ab | 69.3 ab | 70.1 a | 66.3 b |
| **R1** | 300 a | 300 a | 270 b | 240 c | **R1** | 71.6 a | 70.8 a | 68.4 a | 71.5 a |
| **R2** | 298 b | 316 a | 272 c | 241 d | **R2** | 68.9 b | 73.6 a | 67.2 b | 69.7 b |
| **R3** | 304 a | 298 a | 275 b | 243 c | **R3** | 73.0 a | 72.8 a | 69.3 a | 68.4 a |
| **AVG** | **298** | **299** | **271** | **241** | **AVG** | **70.5** | **71.6** | **68.8** | **69.0** |
| | **Pseudostem Height: Circumference Ratio** | | | | | **Third Youngest Leaf, Dimensions and Area** | | | |
| **Cycle** | **Grand Naine** | **Gal** | **Zelig** | **Adi** | | **Grand Naine** | **Gal** | **Zelig** | **Adi** |
| **PC** | 4.23 a | 4.08 a | 3.79 a | 3.59 b | **Length (m)** | 2.32 a | 2.44 a | 2.14 b | 1.93 c |
| **R1** | 4.19 a | 4.24 a | 3.95 a | 3.36 b | **width (m)** | 0.96 a | 0.95 a | 0.88 b | 0.92 ab |
| **R2** | 4.33 a | 4.29 a | 4.05 a | 3.46 b | **area (m$^2$)** | 2.68 a | 2.81 a | 2.28 b | 2.15 b |
| **R3** | 4.16 a | 4.09 a | 3.97 a | 3.55 b | **Length/width ratio** | 2.41 a | 2.55 a | 2.43 a | 2.10 b |
| **AVG** | 4.23 | 4.18 | 3.94 | 3.49 | | | | | |

Length and width of the third youngest leaf were measured after bunch emergence, and leaf area per plant and length-to-width ratio were calculated. As reported by Saúco et al. [16], cultivar height is positively correlated with leaf dimensions. The maximal length of leaf 3 in 'Grand Naine' and 'Gal' was significantly higher than the short-stature cultivars 'Adi' and 'Zelig' (Table 1). Leaf maximal width was identical in 'Grand Naine' and 'Gal', whereas 'Zelig' leaf width was the smallest, and 'Adi' exhibited intermediate width (Table 1). Leaf length-to-width ratio is a useful parameter for discriminating between banana cultivars. 'Grand Naine', 'Gal', and 'Zelig' leaf length-to-width ratios were similar (Table 1) and resembled their respective ratios reported from the Canary Islands [16], whereas the ratio in 'Adi' was lower (Table 1). This suggests that the source of 'Adi' is 'Dwarf Cavendish', whereas the other cultivars are derived from the Cavendish cultivars 'Williams' (Gal) and 'Grand Naine' (Zelig). Despite these differences in leaf dimensions, the area of leaf 3 in the short-stature cultivars 'Adi' and 'Zelig' was significantly lower compared with 'Grand Naine', 20% and 15%, respectively, whereas 'Gal' leaf area per plant was 5% higher than 'Grand Naine' (Table 1). The dimensions and area of the sixth youngest leaf were similar to the dimensions recorded for the third youngest leaf in all cultivars (data not shown).

Correlations between pseudostem height and bunch weight in the R1 crop cycle were calculated. A strong correlation was detected for all cultivars. Interestingly, correlations were higher in the short-stature cultivars 'Zelig' ($r^2 = 0.86$) and 'Adi' ($r^2 = 0.84$), compared with those of the taller cultivars 'Gal' ($r^2 = 0.73$) and 'Grand Naine' ($r^2 = 0.76$).

### 3.2. Reproductive Parameters

In Israel, the cycles consisted of a major parent plant population, with bunch emergence from July–August, and a minor, follower plant population, with bunch emergence from October–November (Figure 1a).

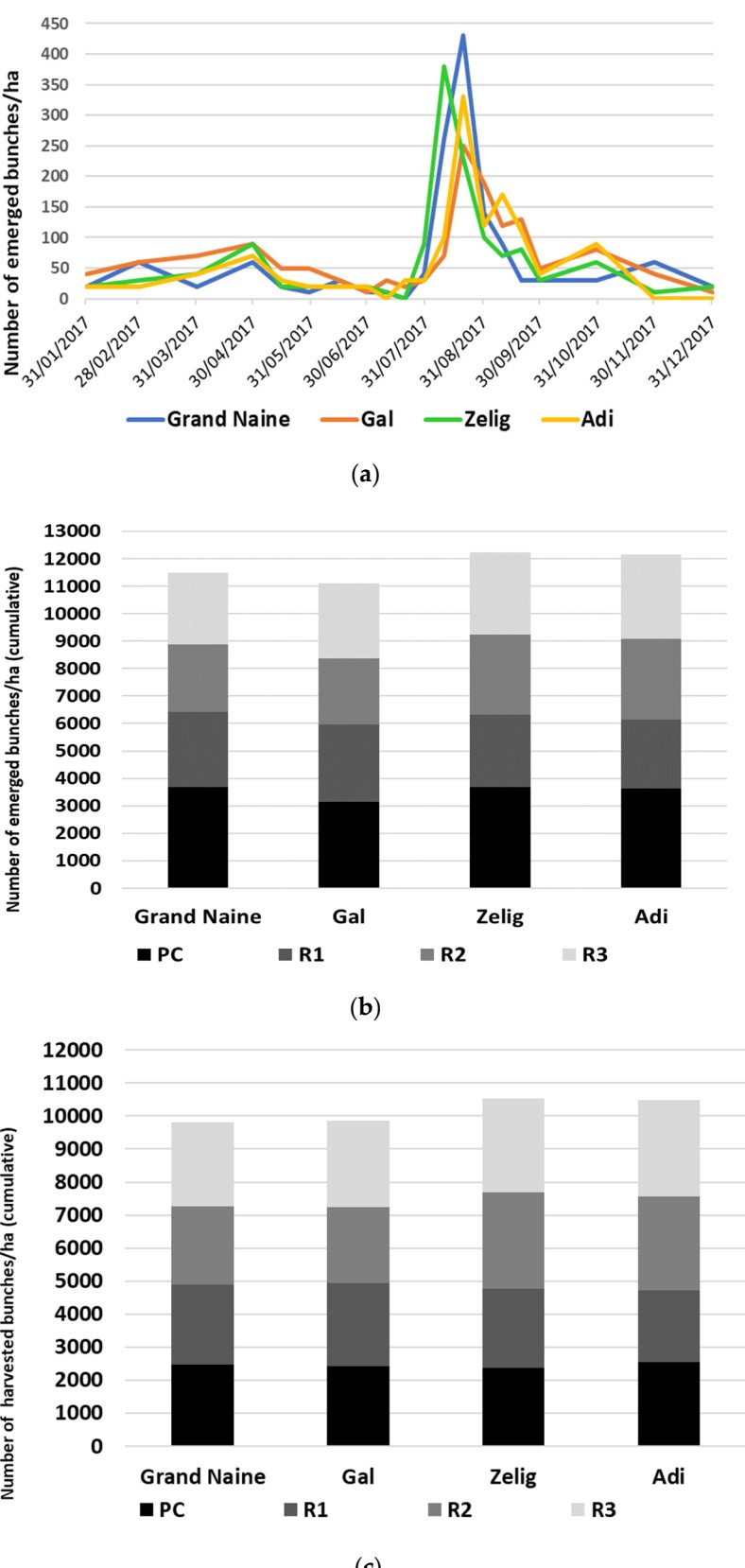

**Figure 1.** Reproductive traits. (**a**) Bunch emergence distribution in cycle R1 expressed in number of bunches on each of the bunch emergence dates (dd/mm/yyyy), (**b**) cumulative number of emerged bunches per hectare over PC–R3 crop cycles, and (**c**) cumulative number of harvested bunches per hectare over PC–R3 crop cycles.

In the PC cycle, the bunch emergence date of cv. 'Adi' was significantly delayed, by 12 days, compared with 'Grand Naine' (Table 2). However, the average delay over the R1–R3 cycles was only 3 days. 'Gal' and 'Zelig' bunch emergence date was the same as that of 'Grand Naine' in the PC cycle, and average bunch emergence date compared with the reference cultivar in the R1–R3 cycles was identical in 'Zelig' and 3 days earlier in 'Gal' (Table 2). Differences in bunch emergence date among the tested cultivars did not result from variation in the proportions of parent plant and follower populations in each of the crop cycles. Figure 1a, depicting bunch emergence distribution in R1, clearly shows earlier bunch emergence of 'Zelig' plants in the major peak from July–August. All cultivars showed a minor, second peak of bunch emergence from October–November, generated by the followers (Figure 1a).

**Table 2.** Reproductive parameters for the various cultivars in each crop cycle and average of all cycles (*n* = 150 plants/cultivar for bunch emergence date and bunch-filling duration in the various cycles). AVG is the mean of R1–R3 (bunch emergence date and bunch-filling parameters) and PC–R3 (emerged bunches per hectare and harvested bunches per hectare) cycles. Values in a row followed by the same letter are not significantly different (*p* > 0.05).

| | Bunch Emergence Date (dd/mm/yy) | | | | | Bunch-Filling Duration (days) | | | |
|---|---|---|---|---|---|---|---|---|---|
| Cycle | Grand Naine | Gal | Zelig | Adi | Cycle | Grand Naine | Gal | Zelig | Adi |
| PC | 06/08/16 b | 06/08/16 b | 07/08/16 b | 18/08/16 a | PC | 144 ab | 143 ab | 133 b | 162 a |
| R1 | 08/08/17 a | 28/07/17 a | 03/08/17 a | 09/08/17 a | R1 | 102 a | 99 a | 106 a | 106 a |
| R2 | 28/07/18 a | 22/07/18 a | 20/07/18 a | 31/07/18 a | R2 | 116 a | 106 a | 113 a | 114 a |
| R3 | 02/08/19 a | 10/08/19 a | 14/08/19 a | 07/08/19 a | R3 | 121 a | 116 a | 117 a | 128 a |
| AVG | 08/02 | 07/30 | 08/02 | 08/05 | AVG | 113 | 107 | 112 | 116 |
| | Emerged Bunches/ha | | | | | Harvested Bunches/ha | | | |
| Cycle | Grand Naine | Gal | Zelig | Adi | Cycle | Grand Naine | Gal | Zelig | Adi |
| PC | 3691 a | 3135 a | 3671 a | 3631 a | PC | 2460 a | 2421 a | 2381 a | 2540 a |
| R1 | 2718 a | 2817 a | 2659 a | 2500 a | R1 | 2421 a | 2520 a | 2381 a | 2180 a |
| R2 | 2480 b | 2420 b | 2896 a | 2956 a | R2 | 2381 b | 2301 b | 2933 a | 2837 a |
| R3 | 2599 b | 2738 b | 2996 a | 3055 a | R3 | 2539 b | 2619 b | 2837 a | 2936 a |
| AVG | 2872 | 2778 | 3056 | 3036 | AVG | 2450 | 2465 | 2633 | 2623 |

Bunch-development rate is expressed as: bunch-filling duration—the number of days between bunch emergence and harvest. In the PC cycle, 'Adi' showed the slowest bunch development, 19 days more than 'Grand Naine', whereas 'Zelig' showed the fastest bunch-filling rate (Table 2). As expected, the following R1–R3 crop cycles were much faster. 'Adi' took slightly longer on average, whereas 'Gal' and 'Zelig' bunch development was 6 days and 1 day faster than 'Grand Naine', respectively (Table 2).

The number of emerged bunches per hectare in cvs. 'Adi' and 'Zelig' in the PC and R1 crop cycles was slightly lower than in 'Grand Naine'. However, in the R2 and R3 cycles, this trend was notably reversed, with the cumulative number of emerged bunches per hectare in the four cycles (12,142 for 'Adi' and 12,222 for 'Zelig') being 5.7% and 6.4% higher, respectively, than in 'Grand Naine' (Table 2, Figure 1b). 'Gal' appeared to be less productive, with a 3.3% lower cumulative number of emerged bunches compared with 'Grand Naine' (Table 2, Figure 1b).

All tested cultivars outperformed 'Gal' in terms of cumulative number of harvested bunches. 'Zelig' ranked first, with 10,592 bunches/ha (7.4% more bunches than 'Grand Naine'), followed by 'Adi' (10,493, 7.1%), 'Gal' (9861, 0.6%), and 'Grand Naine', with 9801 bunches (Table 2, Figure 1c).

### 3.3. Yield Traits

Average bunch weight in the four crop cycles was highest in 'Gal' among the tested cultivars (Table 3). Interestingly, heavy bunch crop cycles (PC and R2) were followed by reduced bunch weight in the following crop cycles (Table 3). 'Adi' presented stable and consistently higher bunch weight than 'Grand Naine' in all crop cycles, whereas 'Zelig' was slightly inferior for all cycles (Table 3).

**Table 3.** Yield parameters for the various cultivars in each crop cycle and average of all cycles ($n$ = 150 plants/cultivar). AVG is the mean of PC–R3 for all traits. Values in a row followed by the same letter are not significantly different ($p > 0.05$).

| | Bunch Weight (kg) | | | | | Yield (ton/ha) | | | |
|---|---|---|---|---|---|---|---|---|---|
| **Cycle** | **Grand Naine** | **Gal** | **Zelig** | **Adi** | **Cycle** | **Grand Naine** | **Gal** | **Zelig** | **Adi** |
| PC | 29.6 b | 33.4 a | 29.0 b | 31.4 ab | PC | 79.5 a | 79.4 a | 80.1 a | 77.4 a |
| R1 | 29.4 a | 29.3 a | 29.4 a | 30.4 a | R1 | 71.1 a | 73.9 a | 69.8 a | 66.2 a |
| R2 | 31.0 bc | 33.6 a | 29.6 c | 31.7 b | R2 | 73.4 b | 77.2 ab | 83.9 ab | 93.2 a |
| R3 | 27.8 a | 29.4 a | 26.8 a | 28.7 a | R3 | 70.8 a | 76.7 a | 75.8 a | 84.2 a |
| **AVG** | **29.5** | **31.4** | **28.7** | **30.6** | **AVG** | **73.6** | **76.7** | **77.2** | **80.2** |
| | Hand Number in Bunch | | | | | Finger Weight (g) | | | |
| **Cycle** | **Grand Naine** | **Gal** | **Zelig** | **Adi** | **Cycle** | **Grand Naine** | **Gal** | **Zelig** | **Adi** |
| PC | 12.3 a | 11.8 a | 11.9 a | 12.1 a | PC | 148 a | 158 ab | 147 ab | 136 b |
| R1 | 11.9 a | 11.8 a | 11.6 a | 12.1 a | R1 | 173 a | 181 a | 177 a | 178 a |
| R2 | 12.6 a | 13.1 a | 12.3 a | 13.0 a | R2 | 177 a | 185 a | 170 a | 161 a |
| R3 | 12.3 a | 12.6 a | 11.7 a | 12.3 a | R3 | 156 a | 153 a | 151 a | 146 a |
| **AVG** | **12.3** | **12.3** | **11.9** | **12.4** | **AVG** | **164** | **169** | **161** | **155** |

Yield in the trial plot was high in all cycles. Accordingly, all tested cultivars outperformed the industry standard cv. 'Grand Naine' in terms of cumulative yield. The highest-yielding cultivar was 'Adi' due to its cumulative yield of 320.1 ton/ha (8.8% higher yield than 'Grand Naine'), followed by 'Zelig' (309.5 ton/ha, 5%) and 'Gal' (307.2 ton/ha, 4.2%) (Figure 2).

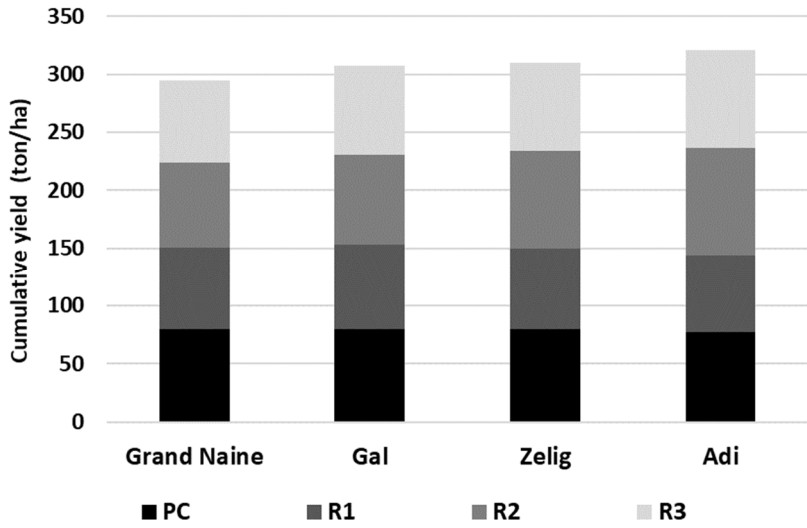

**Figure 2.** Cumulative yield (ton/ha) over the four crop cycles in the various cultivars.

Finger weight is an important trait in banana, directly affecting fruit price. Although only statistically significant in the PC cycle, a trend of lower finger weight was recorded in 'Adi', with an average finger weight reduction of 5.1% compared with 'Grand Naine', whereas 'Gal' fingers were 3.4% heavier than those of 'Grand Naine' (Table 3). Number of hands in the bunch was similar in all tested cultivars, except 'Zelig', which was slightly inferior to 'Grand Naine' (Table 3).

## 4. Discussion

The first standard cultivar in Israel was 'Dwarf Cavendish', which was widely grown in the 1920s. The compact and sturdy plant architecture conferred high wind resistance and facilitated bunch management. Additional advantages of 'Dwarf Cavendish', typical to short-stature banana cultivars, included fast crop cycles and high planting densities [15]. The demand for improved fruit quality, mainly larger fruit size, resulted in the transition to cv. 'Williams', which had ideal bunch and finger characteristics. This was later replaced by cv. 'Grand Naine', combining good fruit quality, faster cycles, and moderate height [22,23].

In light of the statement by Robinson et al. (1993) that "a clone with 'Dwarf Cavendish' stature and 'Williams' bunch characteristics would thus be an ideal banana for the subtropics", the goal of this study was to identify short-stature and high-yielding alternatives to the industry standard cv. 'Grand Naine'. The two cultivars 'Zelig' and 'Adi' were significantly shorter than 'Grand Naine'. However, the current clone of 'Gal' evaluated in this study has lost its short stature due to continuous rounds of selection focusing on higher bunch weight rather than plant height (Gal Or, personal communication). 'Zelig' and 'Gal' were studied in a field trial in the Canary Islands. In agreement with the current study, 'Zelig' had a shorter stature than 'Grand Naine', with a 59 cm reduction in plant height [16]. However, in the latter study, 'Gal' was also shorter than 'Grand Naine', highlighting the evolution of height over the years in cv. 'Gal'.

Annual yield level is determined by a combination of bunch weight and number of harvested bunches per unit area. The four-cycle cumulative yields of all tested selections were higher than that of 'Grand Naine'. Number of harvested bunches was similar for 'Gal' and 'Grand Naine'. Therefore, the increase in cumulative yield can be explained by heavier bunches. The two short-stature cultivars, 'Adi' and 'Zelig', produced more harvested bunches. Although 'Zelig' was the fastest producer, in accordance with the results obtained in the Canary Islands [16], and 'Adi' was the slowest, 'Zelig' did not produce more follower plants (producing additional bunches harvested in the same year) than 'Adi'. Therefore, the higher number of harvested bunches in these short-stature cultivars presumably resulted from their reduced co-shading, which allowed a larger number of plants per unit area in crop cycles R2–R3. These results suggest that 'Adi' and 'Zelig' can be successfully grown at higher planting densities.

Short-stature cultivars are prone to bunch choking and the production of short fingers in the cold winters of the subtropics [15,24]. Indeed, high levels of choke throat (15–20% of the bunches), and inferior bunch and finger weight, 9% and 8% less than 'Williams', respectively, were major drawbacks of 'Dwarf Cavendish' cultivation in Israel, [22]. The average four-cycle bunch-loss rates recorded in the present study in the short-stature cultivars 'Adi' and 'Zelig' (13.5% and 13.8%, respectively) were slightly lower than that for 'Grand Naine' (14.7%), suggesting that no massive bunch choking occurred among the short-stature cultivars in the trial plot. As regards fruit quality, finger weight was slightly and insignificantly lower in 'Zelig' (1.5%) and 'Adi' (5%) compared with 'Grand Naine'.

Water-use efficiency describes the amount of biomass produced per unit of water used by plants. Most of the water loss in the highly shaded banana plantation soil in Israel is attributed to leaf transpiration. Measurements of the third leaf's dimensions revealed a 21% ('Adi') and 12% ('Zelig') reduction in leaf area compared with 'Grand Naine'. The area of leaf 3 was used to calculate leaf area per plant [25]. The significant reduction in leaf 3 area observed in this study suggested a potential decrease in leaf area per plant in cvs. 'Adi' and 'Zelig' compared to the industry standard cv. 'Grand Naine'. Future detailed

quantification of plant leaf area, water consumption, and bunch weight should enable better determination of water-use efficiency in these cultivars.

In the current study, as well as in past studies [9], there was a strong positive correlation between plant height and bunch weight. We found an interesting trend among the tested cultivars: the shorter the cultivar, the stronger the correlation. Thus, the highest correlations were found for 'Adi' and 'Zelig' (0.84 and 0.86, respectively), and the lowest for 'Gal' and 'Grand Naine' (0.73 and 0.76, respectively). Through the use of biotechnological tools, elucidation of the genetic and physiological mechanisms allowing the appearance of short-stature mutants with no reduction in bunch weight should open new opportunities for the development of high-yielding short-stature cultivars.

## 5. Conclusions

The shift to banana cultivation in screenhouses in Israel has resulted in a significant increase in plant height and a consequent increase in production costs. In the present study, the genetic approach to plantation-height reduction was considered. Field evaluation of advanced selections of the local cultivars 'Adi', 'Zelig', and 'Gal', selected for their reduced height, was carried out. While these advanced selections of 'Adi' and 'Zelig' maintained a significant short stature compared with the standard cv. 'Grand Naine', selection for a 'Gal' mutant with bigger bunches resulted in the loss of its short stature. Cumulative yield of all cultivars was higher than that of the industry standard 'Grand Naine', due to higher plant density for 'Adi' and 'Zelig', and heavier bunches for 'Gal'. This multiyear study highlights 'Adi' and 'Zelig' as highly productive short-stature cultivars. The areas of leaves 3 and 6 were reduced in 'Adi' and 'Zelig', suggesting the potential for improved water-use efficiency with cultivation at higher densities and reduced transpiration. A comparative study of water consumption in the low-stature cultivars is ongoing.

**Supplementary Materials:** The following supporting information can be downloaded at: https://www.mdpi.com/article/10.3390/horticulturae8070619/s1, Table S1: Physical properties and mineral content of the soil in the research plot. Samples were collected from the 2–30 cm layer, Table S2: Multiyear meteorological data. Data collected in the Jordan Valley Banana Research Station, Zemach, Israel. Data are means of the years 2000–2020.

**Author Contributions:** I.E.: study design, undertaking of the experiment, data collection, and statistical analysis. A.L., A.R., A.T., G.H.: undertaking of the experiment and data collection. N.'G.: conceptualization of the research, study design, funding acquisition, interpretation of results, and writing of the article. All authors have read and agreed to the published version of the manuscript.

**Funding:** This work was supported by the Israeli Banana Board and Ministry of Agriculture and Rural Development, grant no. 21-55-0001.

**Institutional Review Board Statement:** Not applicable.

**Data Availability Statement:** Not applicable.

**Acknowledgments:** The authors extend their thanks to Emi Lahav and Yair Israeli for fruitful discussions; and to Alex Kaplan and the banana plantation team of Kibbutz Kinneret for trial plot maintenance and assistance with data collection.

**Conflicts of Interest:** The authors declare that they have no known competing financial interest or personal relationships that could appear to influence the work reported in this paper.

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
