# Peer review of "Less Is More? Field Evaluation of Short-Stature Banana Cultivars in a Mediterranean Environment"

_horticulturae, doi:10.3390/horticulturae8070619_

Round 1
Reviewer 1 Report
Interesting new banana varieties with weaker growth. In several places marked in the text information and the results of other studies on banana varieties should be detailed

Author Response
Dear editor and reviewer,
Thank you for reviewing my manuscript, entitled “Less is more? Field evaluation of short-stature banana cultivars in a Mediterranean environment”, and providing valuable comments.
Please find below my responses to the reviewers’ comments.
I did my best to correct in a thorough manner the manuscript (please find attached the revised manuscript) according to the comments.
All the best,
Navot
Line 2:
Reviewer comment: “this content is redundant”
Author reply: I think the warding, “Less is more?” is not redundant. I used it to hint for the essence of the study: the superiority of two of the tested short cultivars compared to the industry standard cultivar. I suggest to keep the original title.
Line 8:
Reviewer comment: “is it about cultivation under cover nets”
Author reply: “under nets” has changed to “in screenhouses” (line 8 in the revised manuscript).
Line 10:
Reviewer comment: “the concept is too general to specify that it is usually about reducing the height of trees
”
Author reply: the genetic approach means that we attempted to reduce plantation height by using cultivars with shorter plants. The next sentence (line 10) introduces the genetic approach.
Line 11:
Reviewer comment: “again the bizarre is just about growth”
Author reply: “vegetative” has been changed to “growth” (line 12)
Line 27:
Reviewer comment: “use the same parentheses everywhere”
Author reply: thank you. Corrected throughout the text
Line 68:
Reviewer comment: “this aspect is very important for the work, it should be discussed in detail”
Author reply: the paragraph has fortified, and new references added (lines 69-76).
Line 107:
Reviewer comment: “it was worthwhile to study the soil's mineral content and compile rainfall and temperature so that other researchers could compare their results”
Author reply: soil physical and chemical properties and multi-year meteorological data added (lines 123-128, and supporting material, Tables S1 and S2, page 14).
Line 110:
Reviewer comment: “whether other care treatments were performed during cultivation, apart from fertilization, e.g. cutting, chemical protection”
Author reply: Clarification regarding chemical application added (line 120-122) this is the only measure used in Israel to control pests and diseases.
Line 164:
Reviewer comment: “Growth”
Author reply: “vegetative” has been changed to “growth” (lines 149 and 183).
Line 283:
Reviewer comment: “give concrete results, all on the level of generalizations”
Author reply:
Concrete data added: “: with 59 cm reduction in plant height” (line 305)
Line 292:
Reviewer comment: “note as above”
Author reply: concrete data added (lines 321-323).
Line 301:
Reviewer comment: “provide in the methodology”
Author reply: description of leaf size calculation appears in line 120.
The sentence “The area of leaf 3 was used to calculate leaf area per plant (23),” has been modified (lines 334-335), to make it clearer to the reader.
Line 326:
Reviewer comment: “in what direction to conduct further research”
Author reply: clarification sentence added (line 360-361).

Reviewer 2 Report
The manuscript entitled "Less is more? Field evaluation of short-stature banana cultivars in a Mediterranean environment" by Idan Elingold et al., provides very useful information regarding the agronomic value of two short-stature bananas cultivars.
The manuscript is very well written and easy to read.
The introduction section provides the appropriate information to the reader in order to acclimate to the scope of the work.
The Material and Methods section is very descriptive regarding the cultivation tasks and measures are taken.
The results are very well presented and the Tables and Graphs are well organized. In the discussion section, all the necessary information is given to support the results.
The manuscript is worthy of being published in its current form.
Author Response
There were no suggestions for modifications.
